# The Interdependence of Autonomous Human-Machine Teams: The Entropy of Teams, But Not Individuals, Advances Science

**DOI:** 10.3390/e21121195

**Published:** 2019-12-05

**Authors:** W. F. Lawless

**Affiliations:** 1Departments of Math & Psychology, Paine College, Augusta, GA 30901, USA; w.lawless@icloud.com; 2NRL Summer Faculty, 4555 Overlook Ave SW, Washington, DC 20375, USA

**Keywords:** interdependence, human-machine teams, entropy, negentropy

## Abstract

*Key concepts*: We review interdependence theory measured by entropic forces, findings in support, and several examples from the field to advance a science of autonomous human-machine teams (A-HMTs) with artificial intelligence (AI). While theory is needed for the advent of autonomous HMTs, social theory is predicated on methodological individualism, a statistical and qualitative science that neither generalizes to human teams nor HMTs. Maximum interdependence in human teams is associated with the performance of the best teams when compared to independent individuals; our research confirmed that the top global oil firms maximize interdependence by minimizing redundant workers, replicated for the top militaries in the world, adding that impaired interdependence is associated with proportionately less freedom, increased corruption, and poorer team performance. We advanced theory by confirming that the maximum interdependence in teams requires intelligence to overcome obstacles to maximum entropy production (MEP; e.g., navigating obstacles while abiding by military rules of engagement requires intelligence). *Approach*: With a case study, we model as harmonic the long-term oscillations driven by two federal agencies in conflict over closing two high-level radioactive waste tanks, ending when citizens recommended closing the tanks. *Results*: While contradicting rational consensus theory, our quasi-Nash equilibrium model generates the information for neutrals to decide; it suggests that HMTs should adopt how harmonic oscillations in free societies regulate human autonomy to improve decisions and social welfare.

## 1. Introduction

To set the stage, humans and machines are evolving into the age of autonomous human-machine teams (A-HMTs) at a time when the need exists to make ever-faster decisions than ordinary humans can process, which motivates the management of complex systems and A-HMTs with AI (e.g., from self-driving vehicles to reducing accidents to hypersonic missiles in combat; in [1,2]). 

The National Transportation Safety Board [3] investigated the death of a pedestrian in 2018 to illustrate with an example of a faulty HMT, finding that the Uber self-driving car saw the pedestrian 6s early, applied the brakes 1.3s early, but the brakes had been disconnected by Uber engineers to improve its ride. The human operator saw the pedestrian 1s early but hit the brakes 1s after impact. However, the Uber car did not update the context for its operator when it could have; its lack of interdependence with its partner made it a poor teammate [1].

### 1.1. Theory. Interdependence

Science has advanced our knowledge of human teams [4]; from them, first, we know that individuals working interdependently in teams are more productive than the same individuals who comprise an independently working team. From Salas’ research team (i.e., [5]), second, we know that when survival is at stake, well-trained human teams function most effectively with “interrelated knowledge, skills and attitudes... [but while teams can function] with great efficiency... poor teamwork can have devastating results” (e.g., plane crashes, friendly fire, surgical complications; [5], p. 279). Third, also from Salas, we know that task-work skills (e.g., a copilot’s duties) should be separated from teamwork skills (e.g., a copilot’s ability to communicate). Our study builds on these three findings; however, in general, very little regarding the science of individuals [6] is transferable to HMTs. We review interdependence theory, prior findings, and new research to understand why. 

### 1.2. Interdependence, Bistability, Measurement and Non-Factorability

Making interdependence a respected subject of study again, the National Academy of Sciences [4] concluded that interdependence is critical in helping teams to perform superior to the same collection of individuals independent of each other, with the best teams being the most interdependent [7]. However, the Academy left what is interdependence open, as was the optimum size of teams, addressed herein. Below, we arbitrarily separate interdependence into these effects: bistability, measurement, and non-factorability [8]. 

#### 1.2.1. Bistability: Two-Sided Stories

First, the bistable information from the construction of a context shared by two or more agents universally generates two-sided stories (e.g., ”he said, she said”). Or, different stories arise from observers and actors [9]. Or, bistable images (Figure 1) and interpretations exist. World views can persist (e.g., religious or atheist, conservative or liberal, myths, or fantasies) when bistable views are sufficiently stable, promoted, or controlled. If sufficiently stable, opposing views can evolve into different tribal customs with supporting functions and hierarchies [10]. Bistable views promote competition among views that forces them to evolve. e.g., “Fracking Buzzwords Evolve, From ‘Ramp Up’ to ‘Capital Discipline’ ” (in [11]). However, few humans like to exchange two-sided or contradictory views, motivating the replacement of disliked views with conformity to a favored view; e.g., humans have made extraordinary advances with quantum theory, yet a consensus regarding its meaning remains elusive a century after its discovery, which is underscored by attempts to supplant it with a one-sided rational theory [12].

*Competition.* Mann [13] identifies several examples of rational choice theory in cooperative situations throughout nature, but he states that the theory breaks down under conflict, as with competition where interdependence theory flourishes. Darwin [10] found that human groups cooperated to offset competition in their “struggle for existence”. Competition leads to the best ideas [14]; it supports Justice Ginsburg’s ([15], p. 3) argument for an “informed assessment of competing interests”. Competition exposes weaknesses, motivating mergers. (e.g., United Technologies and Raytheon announced a merger to improve their ability to innovate for national security [16]). Similarly, competition among bacteria can lead to an “awareness” of a genetic deficiency that forces horizontal gene transfers (mergers) that offer “protection from predators, pathogens, and environmental stress” (in [17]).

*Innovation*: Competition drives innovation [18], but it flounders under the minority control of command decision-making regimes. Consider that China’s R&D expenditures are second in the world to the United States (U.S.) [19]. However, China’s command directed finance (single-sided views), its weak intellectual property protection, and its rampant corruption impede innovation (e.g., [20]). As General M. Hayden, the former Central Intelligence Agency (CIA) and National Security Administration (NSA) chief, stated to his Chinese counterparts on the innovativeness that has so far eluded China: “You can’t get your game to the next level by just stealing our stuff. You’re going to have to innovate” [21]. To better understand the barriers to innovation in China, Taplin [22] reported: 

“Small private-sector firms often only have access to capital through expensive shadow banking channels, and risk that some better connected, state backed firm will make off with their designs— with little recourse.”

For similar reasons, making movies in China has begun to flounder [23]: 

"Tougher government censorship has blocked potential hits and compelled filmmakers to stick with safe formulas that aren’t winning audiences... In the year after Chinese President Xi Jinping put the Communist Party’s propaganda office in charge of regulating films, China’s box-office totals are headed for their first annual decline in at least a decade."

#### 1.2.2. The Measurement Problem

The second effect from interdependence is a measurement problem that we associate with the duality of behavior and imagination, orthogonality, and multi-tasking. This problem commonly negates the value of self-reported questionnaires (e.g., the irrelevance of self-esteem and academics; in [24]; ego-depletion, likely the most famous theory to fail in social psychology; in [25]; and, the mis-informed views of managers about their own organization’s performance; in [26]). The measurement problem is aggravated regarding censoring one world view in favor of another’s, causing individuals, teams, organizations, or the tribes that act on single-minded decisions to be more likely associated with error (e.g., Boeing’s 737 Max accidents; in NTSB [27]; also, in the case study, we review the remediation of the environmental destruction caused by the U.S. Department of Energy’s (DOE’s) mismanagement of radioactive wastes).

The measurement problem reveals social research to be largely inapplicable to the design of autonomous systems (HMTs). The participants in teams can be counted, but measuring weight and mass have little impact on team autonomy; and, the entropy of a team’s configurations can be measured as well as the flow of information and energy that produces teamwork, but is that sufficient? Using the transformation of water to ice as an analogy initially showed promise for a system’s reduction of configurational entropy generalized to HMTs (agreeing with [5]), but it was insufficient, leading us from hurricanes to bio-crystals and living engines to better capture the thermodynamics of teams. However, these methods ignore a chief characteristic of humans: imagination [28]. 

*Questionnaires*. Questionnaires capture beliefs and imaginations well, but not individual behaviors [29]. We can measure the decisions that humans make, but human justifications are often unrelated to decisions (Tversky, in [30]). Likely related, many of the most important findings from social science are based on surveys, but the results have proved to be difficult to replicate [31], which is a problem not exclusive to social science (e.g., in medicine, see [32]; in statistics, see [33]; in physical science, see [34]). We explored the tension between beliefs and behaviors to draw two conclusions: surveys of cognitive beliefs are commonly unrelated to the behaviors that they purport to predetermine (e.g., implicit racism; in [35]); and this failure probably occurs because observation and action are orthogonal [18]. The former impedes the generalization of social theory to HMTs; but, surprisingly, the latter might be good news for social theory. 

*Orthogonality.* Lewin [36], the founder of group dynamics, concluded that interdependence caused a whole (e.g., a group) to be greater than the sum of its parts, which suggests a loss of information from the simple sum of contributions by individual team members (see also [37]). However, one of the earliest theories of interdependence, that a complementarity of opposites (orthogonals) formed the best human relationships, failed to garner statistical support [38]. We conclude that interdependence among humans, the phenomenon that transmits social effects, misleads intuition at human levels that are similar to entanglement at quantum levels, causing a loss of information from measurement. Notice a similarity between interdependence and the quantum [39]: 

Our ordinary intuition regarding physical systems is that, if we know everything about a system, that is, everything that can in principle be known, then we know everything about its parts. However, Einstein explained to Bohr—in quantum mechanics, one can know everything about a system and nothing about its individual parts.

*Correlations.* Correlations present three problems to interdependence theory: First, orthogonal implies independence, but a zero correlation also implies independence. Second, however, for a close relationship, a correlation means a state of interdependence, mutual relationship, or connection between two or more social elements, for which the evidence does not support (e.g., [40]), a conundrum. We often hear that a correlation does not mean causality; but how can we account for causality with interdependence without a correlation? Third, the absence of a correlation in close relationships has been taken to mean the absence of an interdependent relation between orthogonal social objects. However, if two social objects are interdependent, they are causally connected. Interdependence orders two or more social variables. If a pair is in an interdependent relationship that occupies orthogonal roles, the information they collect and self-report, coming from orthogonal roles, has zero correlation by definition. 

Harari ([28], p. 32) claims that humans live in a dual reality, one objective and one imagined. This dual reality includes the roles that humans play. From Shakespeare [41], “All the worlds a stage, And all the men and women merely players … And one man in his time plays many parts.”

To play these many parts requires different interpretations of reality; e.g., a waitress-cook pair working together is interdependently connected, yet each self-reports that each sees the world differently. Another example of orthogonality and the lack of a correlation is found for the carpus collosum, the two independent halves of a brain work seamlessly together like a team, but once surgically severed into two parts, each half of the brain sees and self-reports the world differently [42]. Instead of independent elements, the same statistical effect occurs if the roles of a team are orthogonal (ship’s captain, ship’s engineer, etc.). 

Let us accept that social theory is predicated on the individual; but, teams are comprised of interdependent individuals [4]; and, that a theory of complementarity for interdependent human couples has failed due to of negligible correlations, indicating no relation or independence [18]. However, if we accept that a state of interdependence is proof of a strong correlation, how to resolve this paradox? If team members are interdependent, they are causally connected; e.g., husband-wife; prosecutor-defense attorney; and, pitcher-catcher-outfielder. The conundrum resolves if information is generated in orthogonal roles, if information is collected from individuals, and if only one interpretation of reality can be processed at a time (i.e., individuals are poor at multitasking; in [43]), then interdependence organizes a team of agents into orthogonal roles, but is observed as individuals. 

*Multi-Tasking.* Individuals multi-task poorly [43]; e.g., using a cell-phone while driving a car. In contrast, multitasking is the function of teams [8,44], making teamwork an emergent property that is unequal to the sum of a team’s individual contributors. Forming and operating a team requires multiple streams of communications (verbal, non-verbal) that include the constructive and destructive interference transmitted by interdependence; e.g., destructive interference among humans includes angry debate, even among the best teams [45]; constructive interference includes support for the behaviors that are chosen for a context, for which AI systems, especially machine learning, have been unable to manage so far [46], a problem that must be solved for effective and efficient human-machine teams (see our Special Issue in AI Magazine; [1]). 

We can model orthogonality with the dot product of two eigenfunctions *A* and *B*.
(1)$$<A|B>=\delta_{i,j}$$
where for the Dirac delta, $\delta_{i,j}=1$ if the two eigenfunctions, *A* and *B*, are identical, 0 otherwise. (Using word2vector, the words democracy, freedom, and majority associate; consensus, unity, and sociology associate; but, these two groups do not.)

#### 1.2.3. Non-Factorable Information

Third, the final effect of interdependence is non-factorability (e.g., the intractability of assigning blame in complex legal cases). Humans work with non-factorable information in courtrooms, in investigations, politics, making movies, tradeoffs, etc. The adversarial approach in the courtroom is the method that humans traditionally use to create a context where the truth can be sought or hidden information exposed like biases (e.g., confirmation bias, in [47]). As an example of a trade-off, from signal detection theory, Cohen [48] reported that a "narrow waveform yields a wide spectrum, and a wide waveform yields a narrow spectrum and that the time waveform and frequency spectrum cannot both be made arbitrarily small simultaneously." 

As a tradeoff, we can know the effect of interdependence, as it acts on the whole (say a team), or its parts (the members of a team), but not both simultaneously [18]. That is why constructing a theory of interdependence proved to be “bewildering” ([49], p. 33).

Bohr [9] first mathematically predicted that a tribe adopts a view of reality that is different from a neighboring tribe. What is the evolutionary advantage of having two tribes with different interpretations of reality? From Simon [50], if every story represents a bounded rationality, but if at least two sides exist to every story, then it takes at least two sides to determine social reality, which sets the stage for adversarial or competitive determinations of reality [8,44]. When neutral agents do not exist in an adversarial context, conflict becomes likely [51]. However, in the U.S., the largest political group is the collection of voters known as independents or neutrals [52]. Neutrals serve two functions: first, when political parties attempt to persuade neutrals to join their side, political conflict is moderated; and, second, like a quasi-Nash equilibrium (each side holds opposing, stable, inflexible beliefs), two political parties drive neutrals to process the discordant information that was generated by the opposing parties, while the neutrals justify their votes. An outcome is a political side that wins, a jury’s decision, or the formation of a scientific consensus. Another possible outcome is a compromise. Neither side likes a compromise, but a compromise can last if judged to be fair [53]. 

As a sidebar, Nash’s [54] equilibrium solves the non-cooperative Prisoners Dilemma Game, which is purposively structured to provide a higher payoff between cooperators than competitors, justifying Axelrod’s [55] claim that cooperation is superior to competition. Instead, we elect to see the Nash equilibrium as the only method for resolving non-factorable information (later, we rename this phenomenon as a social harmonic oscillator). Witness that Nash equilibria do not occur in public venues under authoritarian regimes; in contrast, the adversarial competition that is expressed by a Nash equilibrium is the source of justice in the courtroom [56], the best test of a political question [15], the means to the best ideas in speech [14], and the primary route towards innovation [18]. 

Smallman [57] proposed an adversarial approach for the US Navy as a way to reduce accidents at sea by subjecting decisions under uncertainty to a vote. Recently, the adversarial approach solved the Uber self-driving accident that killed a pedestrian in March 2018 [1]. At first, the Police Chief blamed the pedestrian, who was supposedly on drugs at the time. Next, the software that was used by Uber was blamed because the company struggled to average 13 mi before a human observer had to intervene (by way of a comparison, the Waymo self-driving car averaged about 5,600 mi per human intervention). Finally, NTSB [3] blamed the company for disconnecting the car’s emergency breaks, which had correctly sent a signal to apply braking to its disconnected brakes. 

However, how to model the adversarial approach if we cannot explain causality with AI [53]? Part of the problem is the non-factorable effects of interdependence on the social interaction [8,47] making the rational determination of social causality impossible (e.g., the Syrian war; a nasty divorce; and, hostile mergers) (an example of a hostile merger is Barrick’s bid for Newmont’s gold mining operations [58]). As compared to AI machines that are excellent at classification, even if they cannot explain their decisions, (e.g., discovering incorrect prescriptions; in [59]), our goal for this work-in-progress is to seek a foundation for AI to be able to manage aspects of causality to achieve the best social interpretation or social decision by an A-HMT. (e.g., with opposing case-based reasonings; in [60]).

Before ending this introduction to interdependence theory, it is instructive to the reader that Social Psychologists seek to remove the effects of interdependence in experimental data to better replicate their findings ([61], p. 235, by obtaining independent and identically distributed data), as also recommended by engineers [62] and information theorists [63]. We have countered that removing interdependence to amplify the value of individual choices has not only impeded a theory of teams, but it is also perhaps why social science research cannot be replicated [31]. As an analogy, the removal of interdependence from the study of teams is akin to removing the “pesky” quantum effects to better study the atom.

### 1.3. Our Four Prior Findings on Interdependence

We have theorized that bistability, the means to constructing a context shared by a team (e.g., for science teams, see [4]), a measurement problem, and non-factorability characterize interdependence. If true, bistable action-observation information theoretically explains why book knowledge is not efficacious in improving a physical skill rather than the physical training of that skill, which we address in our first finding. Interdependence theory alone explains why the best size of a team is the minimum number of personnel to complete the mission of a team, addressed in the second and third findings; and, in our fourth finding, returning to bistable action-observation information, as derived from Bohr [9], we find that book knowledge and skills knowledge are orthogonal, which suggests an uncertainty principle with the tradeoffs faced by a team when it is confronted by an obstacle to its mission. 

#### 1.3.1. First Finding

We found that a test of air-to-air combat knowledge learned from fighter-jet classroom studies based on the results of an examination of that knowledge led to no association with air-to-air combat win-loss outcomes determined with multiple regressions in a study of USAF single-seat jet aircraft fighter pilots for USAF educators; at the same time, however, the amounts of training received by these fighter pilots led to significant differences in predicting which pilots would win or lose in air-to-air combat strictly due to the amount of highly-specialized air combat maneuver training that the winning pilots had received as compared to the losing pilots. The USAF educators who paid for this study did not like the results and they replicated the study with questions that were loaded in favor of education; this was the data we analyzed for these USAF educators to, again, find no support for their hypothesis (in [8]). 

#### 1.3.2. Second Finding

In our first direct study of interdependence, we considered interdependence to operate analogously to the communication among quantum entangled particles but occurring between two or more social objects to produce a state of total communication from all of the verbal-nonverbal sources of interference possible in a team (versus Shannon’s two-way communication across a channel). We hypothesized that the more redundancy existing in a team, the lower the team’s level of interdependence, thereby adversely affecting a team’s performance by making a team less efficient, which we attributed to destructive interference [8]. With Kullback–Leibler divergence, (where a distribution with no divergence from another is zero; i.e., ln (P/Q) = ln (Q/Q) = ln 1 = 0), we found that the more interdependent were the members of a team, the better that those teams performed (agreeing with findings reported by the National Academy of Sciences on teams; in [4]; also, [7]). We confirmed this theory in a study of the top oil-firms in the world. Specifically, the more freedom that was associated with an oil firm’s home country of record, the less redundancy existed in its oil-firm businesses; e.g., in 2017, Exxon and Sinopec produced approximately the same amount of oil, but Exxon had only about one-eighth the number of employees as Sinopec. 

#### 1.3.3. Third Finding

In a replication of our first study on interdependence, we extended the oil-firm study by collecting the data from the world’s top militaries. Again, we assumed that the more redundant were the members, but now of a military team, the poorer the team would perform. With Kullback–Leibler divergence again, we obtained results that were significantly in agreement with our first study, reaffirming its findings [44]: that is, we found that the more freedom in a nation, the smaller the size of its military measured by its number of military personnel. In addition, we extended our research by hypothesizing and finding that redundancy was significantly associated with the level of corruption in a country, and that the interdependence in a nation’s teams was significantly associated with the individual freedom and the free-market scores of a nation. In our interpretation, team performance suffered, as increasing redundancy reduced interdependence in a team. This reduction was easy to calculate based on the number of redundant members in a team; that is to say, the larger a team that it took to complete the mission of a team, while holding the mission of the differently sized teams constant, the poorer was the performance of a team; the latter contradicted the National Academy, which had speculated that, although the size of teams was an open question, “more hands make light work” (Ch. 1, p. 13, in [4]). Not only had we contradicted the Academy, but we had also found that redundancy was associated with greater perceived levels of corruption in a country, implying that the redundancy from “more hands” existing in a business or military team might be a payoff, which reduces the constructive effects of interdependence. 

#### 1.3.4. Fourth Finding

We also completed a study of innovation in MENA countries (Middle Eastern and North African countries, plus Israel). We found that the average academic levels of schooling in a country, substituting for the intelligence existing in a team, were significantly related to the patents that were produced by a country [18]. We took academic levels to indicate that the more intelligent the students in a country regarding the fields of endeavor that they had engaged in, the better prepared they were to contribute to that field by knowing or suspecting what was missing in the technology indicated by studying the existing patents and vulnerabilities pertinent to a field. 

#### 1.3.5. Discussions. Findings One through Four

*First Finding:* In agreement with the literature [29], Bohr’s [9] idea of a tradeoff between action and observation guided us to find the value of training over book knowledge when the improvement of a dedicated physical skill, in this case air-to-air combat, is the focus. Book knowledge is more pertinent to mental skills, as in the case of developing patents, comprising our fourth finding. 

*Second and Third Findings:* First, we discuss structural fitness. When a team is well-fitted by well-skilled members occupying skill roles that were optimally arranged to execute the tasks a team was designed to prosecute, and then its structural entropy is minimized. For the perfect team, subadditivity occurs, minimizing the entropy generated by the team’s structure, which makes the “whole greater than the sum of its parts” [36]. Accounting for why this has not been discovered heretofore, when these roles are orthogonal to each other (e.g., lead pilot, wingman), the low level of information that is generated by the perfect structure hinders rational explanations for the team’s successes, as the reasons appear to be opaque to outside witnesses and to the insiders in the team itself (see Equation (2)).
(2)$$S_{A,B}\leq S_{A}+S_{B}$$

With $S_{A,B}$ as joint entropy, Equation (2) reflects the structure costs of a team. It indicates that the team with a perfect structure is able to focus more of its energy to maximize its entropy production (MEP) while accomplishing its mission (Equation 5). From Equation (3), a perfect team structure’s entropy minimizes when its degrees of freedom, *dof*, reduce to a minimum, which allows for it to operate as a unit,
(3)$$lim_{n\to N_{team}}log\left(dof\right)=0$$

From Equation (3), as entropy for a well-functioning team’s structure and its degrees of freedom reduce, it acts like when two ions combine to form a molecule, an opening for intelligence to guide a team to shape its structure, so that, once stable, the team can direct MEP to its mission [64,65].

However, second, when a team has poor team fitness, an excited state, “emotion”, is expressed as added structural entropy indicates that a team’s structure is incorrect from the wrong choice of a teammate, from good skills, but misaligned in the wrong position, or from an unfilled role. In that case, and where emotion is siphoning energy from a team’s mission, Equation (4) reflects that a dysfunctional team is more constituted with independent individuals or factions than with interdependent members of a team, proportionately reducing its ability to complete its mission. In the worst case, energy is expended to rip a structure apart. (e.g., the European Union (EU)–Brexit divorce [66]; and the hostile CBS–Viacom merger that cost both firms significantly for two years, only resuming its merger after the leader at CBS was replaced [67]).
(4)$$S_{A,B}\geq S_{A},S_{B}$$

*Fourth Finding:* In discussing the fourth finding, we reconfirmed the adverse impact that corruption in a country had on performance; that is, the more corrupt that a country was perceived to be, the lower that country’s patent productivity. This result implies that by fully educating themselves in a discipline, by engaging in highly competitive endeavors, humans have done very well to exploit the dual reality within which they live. However of great importance to theory, this fourth study contradicts the results from our first study, which found that the training of physical fighter-pilot skills improved those physical combat skills, while educating fighter-pilots had no effect on performance. These are opposing results that we have interpreted to be orthogonal (see Equation (1)) (e.g., representing information vectors, the result of a dot product for eigenfunctions). That is, and as we should have expected from interdependence theory, discovering new patents requires sharper cognitive skills, whereas becoming the best fighter pilot requires well-trained physical skills, which confirms Bohr’s [9] claim of a tradeoff between action and observation uncertainty. 

To summarize these four prior findings, orthogonal roles in a team must be interdependent; redundancy reduces interdependence and increases corruption; however, orthogonal roles produce independent information; combined, reduced structural entropy, and *dof* account for the bewilderment of interdependence; and, entropy is a measurement path for the science of A-HMTs. 

## 2. Materials and Methods: New Research for Human-Machine Teamwork 

We suggest that a team operates as a network of intelligent agents from a catallaxy of exchanges [68] to direct and maintain sufficient free energy or negentropy, *α*, for it to order, organize and manage itself; to survive; and, to solve a problem or perform work that it has designated for itself, or, if part of an organization’s hierarchy, which it has been ordered to act upon (where negentropy is the potential necessary to perform work; reviewed in [69]). Guided by Wissner-Gross & Freer [70], based on the value of intelligence that we have confirmed in our fourth study [18], assume that a limited mental energy landscape exists [71], which places a premium on the use of intelligence in a team for it to be able to direct its negentropy to perform at MEP. We further assume that the mind’s view of its potential negentropy landscape is at least three-fold: To solve the problems that are encumbered upon it, to find vulnerabilities in its opponents, and to maximize its exploitation of *α*. 

At each time step in a team’s work performance, proportional amounts of *α* under its management or control is converted to entropy, while maximum performance occurs at the maximum transformation of *α* into MEP [65], and while teams can be more productive than the individuals who constitute a team [4], we assume that a team must shape its productivity to achieve MEP [64]. However, we do not have an equation yet for teams. Temporarily, we begin with a series of teams in a system that we model, like a series electrical circuit, where an adversarial Nash equilibrium has two competing teams acting together, like a capacitor, *C*, with each side serving to store and release the information needed to induce an audience to consider its ideas by inducing them like an inductor, *L*, the audience processing the information first in one direction then another, the cascades of information oscillating until an event ends. The audience offers social resistance, like a resistor, *R*, where audiences, and their messages and circumstances, pose different levels of resistance. Similar to voltage, which is the change in energy per unit of charge, we assume that we want the potential to represent the change in negenergy per unit of entropy (*dα/dS*) for our model, where
(5)$$\frac{d\alpha }{dt}=\frac{d\alpha }{ds}\frac{ds}{dt}=V_{s}I_{s}$$

For a series circuit, with *i* (i.e., *ds/dt*) (we use lower case symbols to represent change over time) as the information and *di/dt* as its rate of change: (6)$$L\frac{di}{dt}+Ri+\frac{1}{c}\int idt=\alpha$$

To find the natural behavior, we set Equation (6) to zero, and we then propose $I_{0}e^{ft}$ as a solution, with complex frequency, *f*, where $f=f_{r}+j\omega$, and $f_{r}$ the real part of the frequency plotted on the axis of the reals and *j* (where $j=\sqrt{-}1$) plotted on the axis of imaginary effects (the horizontal and vertical axes, respectively, in Figure 2). For natural behavior, we insert $I_{0}e^{ft}$ into Equation (6) and rearrange to get: (7)$$Lf+R+\frac{1}{Cf}=0$$

For the first root locus solution, not knowing the units, we let *R* = 0, *L* = 1 and *C* = 1, giving $f^{2}+1=0$, or, $f_{1,2}=\pm j$, with both points being plotted on Figure 2 to represent the oscillations of two unending conflicts (see the case study below between the U.S. Department of Energy (DOE) and the U.S. Nuclear Regulatory Commission (NRC), where oscillations were occurring, but with no action being taken). Next, when only *R* changes to 1, we get $f^{2}+f+1=0$, giving intermediate solutions that we omit from Figure 2. When *R* changes to 2, we obtain $f^{2}+2f+1=0$, giving $f_{3,4}=-1$, plotted in Figure 2 as points 3 and 4, which we use to represent a compromise. Lastly, when *R* changes to 3, we get $f^{2}+3f+1=0$, giving $f_{5,6}\approx -.4,-2.6$, being plotted on Figure 2 as points 5 and 6 (point 6 is described in the case study below).

## 3. Results

### 3.1. Case Study: Department of Energy (DOE) Citizens Advisory Boards (CABs) 

*Overview*: For this case study, we review the rational choice model; cooperation and striving for consensus at DOE CABs in their decision-making versus those CABs that use majority rules, DOE’s mixed oxide fuel fabrication facility (known as MOX), and a conclusion. 

### 3.2. Case Study: Rational (Consistent) Reality

Rational choice theory attempts to improve the chances of an individual’s survival by rationalizing decisions. Thus, the primary model of decision-making (e.g., evolutionary theory, political science, and military conflicts), emerged from the research of economists and other social scientists as an attempt to improve the decisions of individual humans by making them more consistent and in line with their preferences [72]. However, to work, it only considers reality from an individual’s perspective along with three assumptions. 

For the first assumption, reality is sufficiently determinable by individuals when their decisions are consistent. When consistent, being predicated on the supposition that an individual’s behaviors converge with what its brain sees and choses, the observers of a behavior determine the rational value of an individual’s choice. This predicate permits the observations of behaviors to implicitly determine an individual’s preferences. This model is particularly popular in military circles, where it is known as the perception-action cycle, or the OODA loop (i.e., Observe-Orient-Decide-Act).

Second, it is assumed that selected individuals (scientists) can train others to observe the consistency of the targeted behaviors. Further, observers sufficiently trained are able to see through deceptions and misguided beliefs. Predicated on the ability to collect the readily available information, when the data for trained observers converge in the aggregate, their analysis forms a consensus by the trained observers in their forecast of the decisions a collective makes.

Third, at the level of a collective, it is assumed that any consensus by a collective should produce its best decisions, known as the rational consensus decision model designed to build consistency for group decisions [73]. Specifically, Mann’s [13] model is “based on perfectly rational individual decisions by identical individuals … ” He concludes: 

“rational decision making is consistent with empirical observations of collective behavior... with individuals demonstrating a strong preference for an option chosen by others, which increases with the number of others who have selected it …”

#### Criticisms

We critique these three assumptions of the rational choice model. First, when data for individual perceptions and actions converge, they may not converge to an actual phenomenon, but a mathematical illusion [74,75]; viz., the brain’s motor and sensory systems are independent [76]. A dual brain system that is composed of independent parts can form beliefs that are inconsistent with behavior. As examples: Habits disattend to behaviors [77]. “Denial is a characteristic distortion in thinking experienced by people with alcoholism” [78]. A mostly forgotten myth held that ghost dances would protect Native Americans from white settlers before many were massacred at Wounded Knee [79]. A current myth is that interdisciplinary science teams provide better science [7]. Simply put, the human imagination can create stories and myths that are disconnected from reality [28].

Second, if social reality can form an illusion, even to trained observers [80], it may explain Tetlock’s belief that he could train humans to become the rational, consistent superforecasters he described in his new book [81]. Yet, after careful selection and strenuous training, on his website, (Tetlock’s data from his "good judgment" website has been deleted and revised; see [8]). Tetlock posted the first of two predictions for 2016 that were made by his personally trained forecasters, with both failing: that Brexit would not occur and that Trump would not be elected President of the United States. We counter that interdependence in close votes generates limit cycles, but not entropy [82].

Worse, deception requires an illusion of consistency, which “is hard to explain” from a rational perspective [83]. For example, a successful technique for confidential informants is to behave in front of those they are informing upon to appear to be consistent with their past behavior (e.g., confidential informants may “preserve their anonymity” if they do not testify in court [84]). Deception is especially critical to enact, or to uncover, in military matters (Sun Tzu, in [85]; for AI and deception in military matters, see [86]); politics [87]; governing [88]; leadership [89]; and, close relationships [90]. Deception is not only difficult to uncover in real life [91], but also in the laboratory (reviewed in [92]). Deception can occur, even among members of the U.S. Federal Reserve (e.g., Teitelbaum [93] reported, “Accusations of furtive elections, opaque decision-making, and privileged access haunt the most prominent outpost of the Federal Reserve”). When deceivers mimic the orthogonal roles expected of them, deception and related behaviors (magic, acting, confabulation) easily fit into the roles that humans adopt and play. Unlike rational choice theory, competition can theoretically uncover deception in social reality (e.g., establishing validity with scientific experiments, in [94]; searching for truth in the courtroom, in [56]; and, experimenting with new naval technology, in [95]).

Third, Kelley [96] found that, in the laboratory, the preferences determined by questionnaires for an individual when alone do not match what an individual choses in a social (game) situation, possibly the rational choice theory’s motivation to impute values based strictly on observations. For the prisoners dilemma game (PDG), no matter what Kelley tried in the laboratory for identifying the strongest preferences of individuals, interdependence transformed the actual choices that they made, impeding game theory from scaling to teams or larger collectives [8,44].

Based strictly on repeated PDG game contests, Axelrod ([55], p. 7–8; also [97]) concluded that competition reduced social welfare: “the pursuit of self-interest by each [participant] leads to a poor outcome for all”. This outcome, he argued, can be avoided with punishment that is sufficient to discourage competition. However, Rand and Novak [98] tried, but failed, to extend Axelrod’s finding from the laboratory to the outside world (p. 413). Rand and Nowak concluded with their hope that the “evidence of mechanisms for the evolution of cooperation in laboratory experiments … [would be found in] real-world field settings” (p. 422). 

The phenomenon of interdependence accounts for all of these differences. Conflict and innovation are central to interdependence theory [18]. Unlike rational choice and game theories, interdependence in the form of an emergence scales to nations [8,44]. From Friedman [99], “Foreign policy is a ruthless and unsentimental process” where: 

“crises are common... The U.S. had a civil war in the 1860s but by 1900 was producing half of the manufactured goods in the world while boasting a navy second only to the British. Internal crises do not necessarily mean national decline. They can mean strategic emergence.”

Kahneman (interviewed by Workman [100]) concluded that the rational choice model does not apply to the average person. We agree. However, Mann’s [13] “model predicts that the decision-making process is context specific” to laboratory contexts less clearly observed in more natural conditions, where his model predicts a more gradual decline in the degree of consensus achieved as the magnitude of conflict is increased. Conflict disables rational choice, but it is central to interdependence theory, in that humans that are freely able to self-organize make decisions in the field by seeking the best arguments judged by majority rule, namely, like in our case study.

### 3.3. Case Study: Department of Energy (DOE) Real-World Decisions

In the real world provided by the U.S. Department of Energy’s (DOE) massive cleanup of the extraordinary radioactive waste contamination, its mis-management spread across the United States [101], at a total cost in billions of dollars for the cleanup of only its Hanford, WA, and Savannah River Site (SRS), SC, DOE’s two sites with the largest inventory of military radioactive wastes, DOE’s cleanup provides a case study of rational consensus-seeking rules found in authoritarian and more socialist countries when compared with traditional majority rules that are found in democracies. 

Background. As a Federal agency, DOE mostly operated as an authoritarian organization overseen only by the National Academy of Sciences and the U.S. Congress [101]. During this time, widespread environmental damage occurred under practices that were permitted by its management, as exemplified by Figure 3. However, after this damage became public knowledge DOE was forced by Congress to include the public in its decision-making, leading to citizen advisors and our case study. 

DOE Citizen Advisory Boards (CABs) are located at major DOE cleanup sites [101]. DOE recommended that its CABs use consensus rules to reduce the conflict from majority rules; however, DOE did not insist, setting up a natural field experiment. The Hanford CAB (HAB) uses consensus-seeking rules and the Savannah River Site CAB (SRS-CAB) uses majority rules. Bradbury and Branch [102] were assigned to evaluate the nine DOE CABs then existing across the United States. From their perspective, Bradbury and Branch saw success as limited to decision processes, but indifferent to the actual cleanup results. From Bradbury and Branch ([102], p. 7), about consensus-seeking, in their view, “best” would include the broadest possible participation, so that “the process of striving for consensus both reinforced and demonstrated members’ commitment to the essential goal of providing advice to DOE and the regulators that would have broad-based support”. (p. 7) Thus, no matter the consequences of the advice rendered, Bradbury and Branch did not favor HAB’s consensus-seeking deliberations, but not the majority rules used by SRS-CAB. 

However, in addition, Bradbury and Branch ([102], Appendix: HAB, p. 12) found at HAB a: Lack of civility, and indulgence in personal attacks during board meetings can erode personal relationships and reduce the effectiveness of board deliberations. Despite a variety of efforts, the board has not managed to adequately control this problem.

Additionally, from Bradbury and Branch ([102], Appendix: SRS-CAB, p. 12), they reported that A shared sense of purpose, pride in the board, camaraderie, and sense of family were very evident.

The “lack of civility” that was experienced by the members of HAB towards each other versus a shared “sense of pride” by the members of the SRS-CAB might alone satisfy our hypothesis for the difference in consensus-seeking rules versus majority rules; instead, we attribute it to the accomplishments in the field driven by these two vastly different styles. One of the internal complaints often heard from HAB members was their inability to make recommendations to DOE at Hanford for the concrete actions that Hanford should take for improving or accelerating its cleanup; in contrast, most of the recommendations by SRS-CAB asked DOE to “accelerate” its cleanup (e.g., [103]; see Table 1 below). When considering the few positive results that were achieved by HAB versus the significant successes achieved by SRS-CAB, we conclude that the conflict that was generated from majority rule achieved a wider consensus, but with better results in the field, than purposively seeking consensus. 

Our colleagues in Japan, which is a nation that favors consensus-seeking, ran a study with college students that were participating in simulated workshops that discussed siting a proposed HLW repository in Japan to contrast consensus-seeking and majority rules. The workshop began with a lecture by a government expert regarding the proposed repository, and participants were then divided into consensus and majority rule groups. The conclusion was that the majority rule participants preferred action versus consensus-ruled participants, they were problem solvers exploring all potential solutions versus problem finders who preferred to foreclose alternatives, and, as occurred with the SRS-CAB, a consensus between citizens and government was more likely with majority than consensus seeking rules [108]. Consensus-seeking rules control a majority by blocking the exploration for the best ideas, solutions, and decisions; i.e., they impede MEP. 

### 3.4. Case Study: A Social Harmonic Oscillator 

Returning to our case study, when compared to HAB, there are too many successes by the SRS-CAB to mention (e.g., it recommended that SRS be the first DOE site to close its seepage basins; to start up the vitrification of HLW in its Defense Waste Processing Facility; to store vitrified HLW at SRS; and, to make the definition of HLW dependent on actual risk). Two successes at SRS, which are relevant for this study, regard the HLW tanks that the SRS-CAB recommended to close. It made the first recommendation to initiate HLW tank closure in 1996 [109]. The first two HLW tanks, tanks 20 and 17, were closed in 1997, the first closed under regulatory authority in the USA and possibly in the world. Subsequently when SRS, supported by the SRS-CAB [110], began to close HLW tank 19 in 2000, DOE was sued to cease from closing additional tanks; and, DOE lost and ceased tank closures. That changed when the National Defense Authorization Act (NDAA) for the fiscal year 2005 became law. It allowed for DOE to resume closing its high-level waste tanks. The SRS-CAB [111] immediately recommended that DOE close both HLW tanks 18 and 19, being strongly supported by DOE and the State of South Carolina.

However, neither SRS-CAB, DOE, nor the State of South Carolina, anticipated what would happen next. The NDAA-2005 law had given the U.S. Nuclear Regulatory Commission (NRC) limited oversight over all future HLW tank closures by DOE; DOE had to gain permission from the NRC to proceed. Subsequently, DOE would propose a plan followed by request from NRC for more information, but with no action taken on HLW tank closures for almost seven years (like a harmonic oscillator, points 1 and 2 in Figure 2). At a public meeting that was held by the SRS-CAB in November 2011, environmental regulators from the State of South Carolina complained in public that DOE was going to miss its legally mandated milestone to close HLW tank 19. After strenuous public debate, the full SRS-CAB [112] recommended that DOE immediately close HLW tanks 18 and 19. 

DOE, the State of South Carolina, and DOE-Headquarters quickly agreed with the SRS-CAB. Once the DOE-CAB and the public got involved, all of the agencies, including the NRC, quickly agreed to DOE’s closure plan. The tanks were closed in what one ranking DOE official described as “…the fastest action I have witnessed by DOE-Headquarters in my many years of service with DOE” (from [101]; reflected by point 6 in Figure 2). 

Since that day, the site has celebrated the 20th anniversary of its first tank closures [113]: 

"Closure is the final chapter in the life of an SRS tank. Once workers remove the radioactive liquid waste from the tank, they fill it with a cement-like grout, which provides long-term stabilization of the tank and ensures the safety of the community and environment surrounding SRS. The first waste tank closure in the nation—Tank 20 at SRS—came about six months before the Tank 17 closure. Each tank held about 1.3 million gallons and began receiving waste from the nation’s defense efforts in 1961. Each SRS tank contains a different combination of insoluble solids, salts, and liquids, which makes each closure unique."

### 3.5. Case Study: Department of Energy (DOE) Mixed Oxide Fuel Fabrication Facility

Next, we briefly review a project by DOE at SRS also with oversight by NRC that failed after 12 years, which was likely from a lack of public scrutiny by SRS-CAB with its persistence to accelerate cleanup (thus, repeating points 1 and 2 in Figure 2). In 2001, DOE signed a contract with Duke COGEMA Stone & Webster for a Mixed Oxide (MOX) Fuel Fabrication Facility. However, DOE kept SRS-CAB from addressing the MOX project.

The facility was being built to take surplus weapon-grade plutonium and mix it with uranium oxide to form MOX fuel pellets for reactor fuel assemblies. Being irradiated in commercial nuclear power reactors, the spent fuel would have contained plutonium in a form less usable for nuclear weapons. However, on November 1, 2018, the White House (DOE) terminated MOX. From World Nuclear News [114],

“Work started on the MOX Fuel Fabrication Facility (MFFF) in 2007, with a 2016 start-up envisaged. Although based on France’s Melox MOX facility, the US project has presented many first-of-a-kind challenges and in 2012 the US Government Accountability Office suggested it would likely not start up before 2019 and cost at least USD7.7 billion, far above original estimate of USD4.9 billion … About 70% complete, the facility was intended to dispose of 34 tonnes of weapons-grade plutonium by turning it into fuel for commercial nuclear reactors”.

### 3.6. Case Study: Summary. Department of Energy (DOE). Minority Control Versus Free Speech

We have identified consensus-seeking as minority control, the reason it is preferred by authoritarians [101]. The European Union reached the same conclusion in a White Paper ([115], p. 29): 

“The requirement for consensus in the European Council often holds policy-making hostage to national interests in areas that Council could and should decide by a qualified majority.”

In summary, the consensus-seeking decision rules employed by HAB have impeded the cleanup of DOE’s Hanford facility in the State of Washington. In contrast, the majority rules that were used by the SRS-CAB accelerated the cleanup at DOE’s SRS site in the State of South Carolina. Of particular interest, the MOX project failed when DOE prevented its SRS-CAB from overseeing the construction of its MOX facility at SRS. Thus, based on interdependence theory, majority rule decisions by citizens works like free speech—the best ideas win, they become the best decisions for taking concrete actions, and the results provide the best social welfare possible. 

## 4. Discussion. Adversarial Governance (Social Harmonic Oscillators) for HMTs

Instead of a rational approach, a social harmonic oscillator is a better model, not for making predictions, but for reaching better decisions, uncovering deceptions, and reducing accidents (Section 4.1). From Mann [13], a rational consensus begins to break apart with conflict, where interdependence theory thrives. If social reality cannot be rationalized, then we expect that the best interpretations and decisions arise from the interdependence that was established by the opposing sides of what we formerly named a quasi-Nash equilibria, as in the adversarial two-party system offered in the U.S.; or, the competition for merger targets and splits forced by competition in free markets ([66]; (e.g., AT&T might split off DirecTV; in [116]) or the adversarial prosecutor-defense attorney model of the courtroom that best leads to justice [56].

Adversarial decision-making advanced the clean-up of DOE’s military radioactive waste mis-management at DOE sites across the U.S., especially at SRS, rather than Hanford. It produced a superior cleanup when compared to the consensus-seeking rules that a minority uses to control the majority that it governs (e.g., autocracy; socialism). From the case study, when decisions were made by seeking consensus, few alternatives were discussed, few solutions were offered, and most concrete decisions were blocked, suggesting wasted entropy production (WEP); however, when decisions were made under majority rule, a greater exploration of potential solutions were discussed and rapid actions taken, which implied MEP. We believe this offers guidance for autonomous HMTs, especially to reduce human accidents (see Section 4.1).

The issue regarding limits of authority from checks and balances introduces discussion about the ethics of autonomous vehicles in use by the military in what are known as humans observing “on-the-loop” autonomous machines, but with humans not directing their decisions; these more fully autonomous operations carry significant risks. On the positive side, self-directed machines may save more lives since most accidents are caused by human error [1]. However, an editorial in the *New York Times* [117] articulated the concerns that AI systems can be hacked, suffer data breaches, and lose control to adversaries. The Editors quoted the UN Secretary General, Antonio Guterres, that autonomous “machines with the power and discretion to take lives without human involvement... should be prohibited by international law”. The editorial in the *New York Times* recommended “humans never completely surrender life and decision choices in combat to machines.

However, given this cautionary note, how should fully autonomous HMTs make decisions? Retired Gen. Mattis [118] wrote that tribal factions are tearing our democracy apart. However, Madison wrote that the U.S. is not a democracy, but a Republic governed by checks and balances that limits power by pitting factions against factions. As we move into the age of intelligent, autonomous HMTs, the key to limiting machine autonomy may not be a rational approach for decision-making, but to use the techniques that humans use to limit the ambitions of factions by limiting their autonomy or by checking them with the ambitions of their orthogonal opponents ([119]; e.g., McLemore and Clark [120] want to limit the autonomy of machines by holding their human authorities responsible).

Consider the value of checks and balances as more than a means of governance, but also as a check on autonomy that also becomes a source of Shannon information (Eqn. 4) available for benefitting society and improving its decisions for those able to act. For example, a recent court case dealt with the reduction in checks and balances when the U.S. Congress defers to the single authority of a Federal agency instead of using competing interests among factions in the Congress to shape U.S. policy (Judge Willett [121]). The competition afforded by checks and balances might be superfluous if humans were able to simply perceive and act. Instead, individual self-awareness is not only bounded [50], but also unable to determine social reality isolated from interdependence, why autocrats and gangs censor and prevent the formation of quasi Nash equilibria [8], which we have modeled as social harmonic oscillators (SHOs).

Not everyone agrees, completely. In an interview, Kahneman (in [100]) stated: “so I would very much like to see the reply and rejoinder way of dealing with [academic] debates disappear”, preferring rules to force adversaries to collaborate and listen to each other without the anger that is displayed when critiques are in play. However, in our model, two-sided debates in front of an audience of neutrals not only interdependently processes the Shannon information that the Nash equilibria generate and convert into a decision, but it also entertains and moderates emotional conflict (based on polling data [52], most individuals in the U.S. electorate self-identify as an independent, making them consistently inconsistent).

Under majority rule, when free speech is openly permitted, the best ideas, concepts and beliefs win [14], leading Justice Ginsburg ([15], p. 3) to argue for an “informed assessment of competing interests”. Scaled up, her argument explains how competing interests use trial-and-error processes in free markets to improve social welfare by satisfying demand [122], whereas the rational approach that is used in command decision economies under minority control cannot ([68]; e.g., socialism in Cuba).

### 4.1. Accidents

Humans cause most accidents [1], which is an intractable problem for the rational perspective. A-HMTs may be able to help to reduce accidents. Once a machine in an A-HMT knows what it is supposed to do, it also knows what its human teammate is supposed to do; a machine so taught can quickly step in when an engineer becomes distracted as a train speeds into a turn (NTSB [123]). Our research team has made a bet that, within five years from 2019, an autonomous machine will prevent an accident by taking control from a misbehaving human operator [124]. For example, a commercial airline pilot might have been prevented from committing suicide had the airplane placed itself into a safe flying mode until the authorities on the ground could have taken control; in 2015, this action, for which the technology exists, would have saved the 150 humans aboard Germanwings flight 9525; in 2001, if the submarine itself had been able to cross-check the decision of its Captain who wanted to demonstrate to his visitors aboard a US Navy submarine a rapid ascent from the ocean floor, it might have saved the lives of the nine Japanese students killed aboard the Ehime Maru, a Japanese fishing vessel, and the US officer’s career [125]; and, in 2018, a signal from the Uber car itself shared with its human operator that the car had detected a change in context 5 sec. before the operator made the detection might have kept the first pedestrian killed by a self-driving car alive [3].

### 4.2. Leadership

Briefly, for now, once a mission has been determined, a team operates best under a single leader’s command until an adjustment is needed [64], the mission has changed, or uncertainty arises. The team should debate its alternatives when tradeoffs are needed, if time permits.

## 5. Conclusions

We assume that teams are free to choose their own members, but, that freedom, especially for successful teams, motivates constructive and destructive interference (entropy). Competition is an example of constructive interference serving to obtain and drive the best teammates (e.g., [7]) to achieve the best team performance (MEP). For destructive interference, gangs, authoritarian rulers, and the decision criteria for minority control (consensus-seeking) impede a team’s performance [101].

Negentropy is required to operate a team to perform work [69]. Two teams competing against each other at a quasi-Nash equilibrium consume the most negentropy to produce MEP [64]. This quasi-Nash equilibrium could reflect a courtroom trial between an equally competent prosecutor and defense attorney [56], between two successful businesses competing to merge with a third, (e.g., the merger duel for Anadarko won by Occidental Petroleum Co. against Chevron [126]) or the recommendations by citizens advising the Department of Energy (DOE) on the cleanup of DOE’s radioactive waste mismanagement at one of its sites, as presented in the case study.

Interdependence suggests that a team forms a network across the team that passes along and shapes negentropy, which is converted into entropy at each step as a team works to fulfill its mission. However, like the measurement problem at the quantum level, the collapse of interdependence during measurement impedes the reconstruction of the action from what is measured. That recognition led us to our new model and our case study of a social harmonic oscillator (SHO).

The operation of teams has long been a mystery, but not their goal of achieving MEP. Ben-Yosef reaffirmed the value of this goal with his recent discovery of the ancient kingdom of the Edomites. “The efficiency of the copper industry in the region was increasing. The Edomites developed precise working protocols that allowed them to produce a very large amount of copper with minimum energy” [127].

Interdependence not only explains Smith’s [122] invisible hand; Lewin’s [36] “the whole is greater than the sum of its parts”; that stable structures are needed to shape the maximum production of entropy; that intelligence is needed to shape teams [64] and navigate obstacles (e.g., rules of engagement) [18]; but also, intelligence is needed to find solutions to problems. However, we have also reported that the whole can be significantly less productive than the sum of its parts ([18]; e.g., a divorce, dysfunctional merger, poorly operated alliance; or, gang, authoritarian, or monopoly control). Thus, how the duties and roles for an A-HMT’s structure are assigned is critical [64].

The theory of interdependence that we have proposed also allows for us to redress social theory.

For example, there are several problems with rational choice theory; scientists, humans possibly with biases who have favored cooperation, cognitively determine the assigned values [8]; but, cooperation has only been proven to be socially beneficial in laboratory experiments, not in the field ([98], p. 422). Rational choice theory is more likely to reach WEP than MEP with minority control versus majority control. We conclude that rational choice theory struggles to scale to collective decision-making, breaking down with conflict, where interdependence theory thrives (e.g., [13]). Thus, we conclude that rational theory does not generalize to A-HMTs, which only an interdependence among competing orthogonal systems can best determine social reality to achieve MEP [1].

Assuredly, our model is temporary, but it advances the theory of autonomous human-machine teams; it integrates wide swaths of evidence from the field. Theoretically, it uncovers the mystery of correlations for orthogonal relationships; and, as applied in our case study, it reconfirms the value of adversarial decision-making and it elevates the potential value of a social harmonic oscillator. In the future, we want to consider replacing our “circuit” model with an Ising quantum model for neutral audiences (to model spins up and down until acted upon by outside social forces) and with Grover’s quantum search algorithm to find those who have been won to one side or the other.

Finally, for our promised surprise to Social Scientists, our model accounts for why surveys of cognitive beliefs are commonly unrelated to the behaviors that they purport to determine (e.g., implicit racism; in [35]); and this probably occurs because cognition and action are orthogonal [18]. The former impedes the generalization of social theory to A-HMTs; but, and surprisingly to us, we have also suggested that the latter might be good for social theory. Today, social science has been experiencing a replication crisis (e.g., [31]), and it is currently struggling to survive as a science. One of its leaders who has encouraged replications, Nosek himself is involved in a similar crisis with implicit racism, determined by a questionnaire that has become one of the biggest money making offerings in all of social science [128]. The good news is that, if implicit bias is negated at the individual level by being produced from orthogonal effects, its real effects may still exist at larger scales, of say towns or states, which has been reported [129].

## Figures and Tables

**Figure 1 entropy-21-01195-f001:**
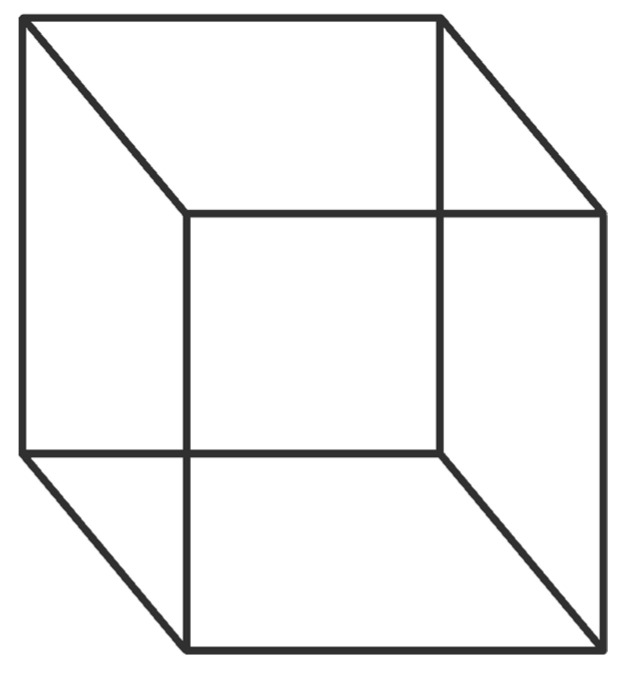
Nekker cube. An observer can see the cube opening either in one direction or the other, but not both simultaneously. One immensurable interpretation could represent conservatives versus liberals; Bohr’s Copenhagen interpretation of quantum mechanics versus Bohm’s pilot wave; or, one tribe’s football club versus another.

**Figure 2 entropy-21-01195-f002:**
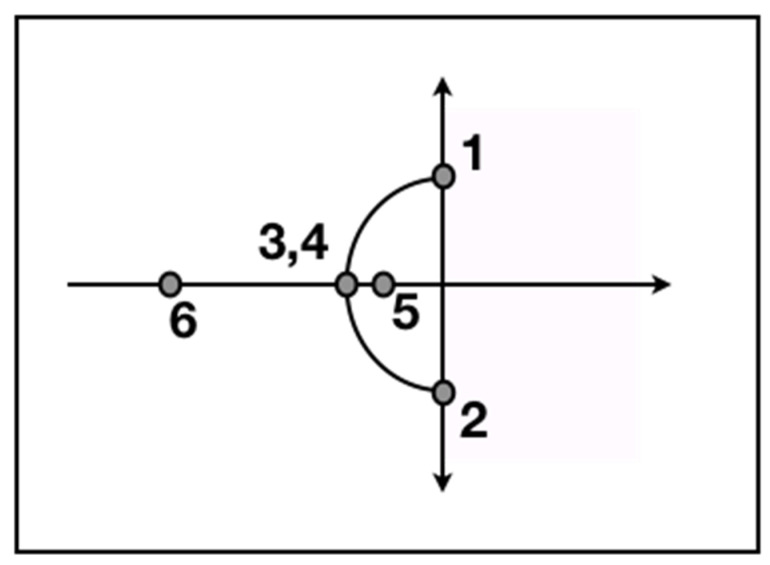
The root locus of a team’s network (for the natural frequency, we focus on energy dissipation to the left of the center point, with increasing energy to the right, ignored in this study). For Equation (6), in cases when no resistance exists (no audience; or two equal sides of an issue talking past each other), solutions fall only on the *y* or imaginary axis, similar to harmonic oscillations (points 1 and 2). When two equal competitors in a debate fully engage with each other before an audience with equal resistance to each, a compromise provides a solution where the *y* curve meets at the *x* or real axis (e.g., points 3,4). When the audience selects one argument to dominate the other, the root locus falls on the *x* axis without oscillations (solution 6 represents strong resistance).

**Figure 3 entropy-21-01195-f003:**
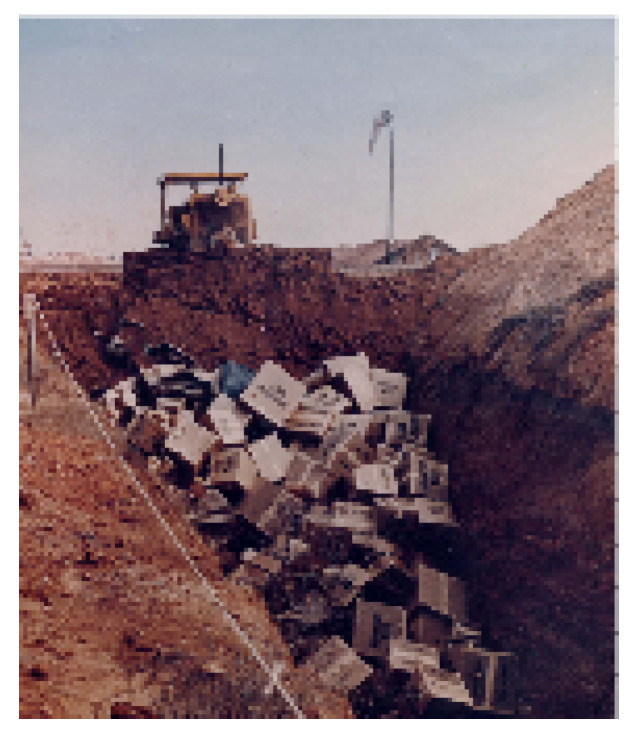
An example of one of the inferior waste management practices by the U.S. Department of Energy prior to 1985 associated with the widespread dispersal of radioactive waste contamination on and off the Savannah River Site, SC; similar mismanagement practices occurred at Hanford, WA. Reflective of this photo, 95% of all of Department of Energy’s (DOE’s) solid military radioactive wastes were disposed in open soil pits inside of cardboard boxes; they often sat in the pits uncovered for months at a time during all weather conditions (public photo by DOE).

**Table 1 entropy-21-01195-t001:** Contrasting the cleanup at Hanford Citizens Advisory Board (HAB) versus that at Savannah River Site (SRS) [104,105,106,107].

Major Site Activity Status	Hanford	SRS
The number of HLW tanks cleaned and closed	0	8 cleaned and closed; from 1997
Canisters poured of vitrified HLW glass and stored ready for shipment to a HLW repository	0	4200 canisters poured and stored; from 1996
Legacy transuranic wastes remaining onsite to be shipped to the repository at WIPP, NM	11,000 drums	750 cubic meters

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
