# Peer review of "The Interdependence of Autonomous Human-Machine Teams: The Entropy of Teams, But Not Individuals, Advances Science"

_entropy, 2019, doi:10.3390/e21121195_

Round 1

Reviewer 1 Report

It is a novel approach to interdependence. The article gives new insights and perhaps a new model.

Author Response

Comments

Replies

Does the introduction provide sufficient background and include all relevant references? Yes.

Thank you.

Is the research design appropriate? Can be improved.

I have re-worked the manuscript to improve the research design.

Are the methods adequately described? Yes.

Thank you.

Are the results clearly presented? Yes.

Thank you.

Are the conclusions supported by the results? Yes.

Thank you.

It is a novel approach to interdependence. The article gives new insights and perhaps a new model.

Thank you.

Reviewer 2 Report

Dear authors.
Thank you for the chance to contribute to your work.
Here are some considerations for improving the paper.

Abstract

The abstract is very long and conceptual. Key concepts should be presented, approaches that have been employed, and the expected results to prompt reading continuity. The current formatting makes reading tiresome.

In the introduction, the word Overview is unnecessary.
The final chapter in the introduction to the paper organization is missing.
The introduction and its subtopics got very confusing. It is not known where the introduction to the theme is being made and where is the theoretical framework. The reading is confusing.

Fix typos and formatting on paper.

The symbols of the equations in the paper should be explained. Some of them were not correctly posted.

Honestly, I had a hard time understanding what the main contribution of the paper was. The massive reading of the texts,
Lack of schematic images impairs the reader's perception.

The article needs to be better explained and highlighted to make it easier for the journal to read interested parties.

Author Response

Comments

English language and style are fine/minor spell check required

I have carefully reviewed the manuscript for spell checking.

Does the introduction provide sufficient background and include all relevant references? Must be improved.

I have edited the manuscript to improve the background and references.

Is the research design appropriate? Must be improved.

I have edited the manuscript to improve its research design.

Are the methods adequately described? Must be improved.

I have edited the manuscript to improve its description of the methods.

Are the results clearly presented? Must be improved.

I have edited the manuscript to improve the clarity of the results.

Are the conclusions supported by the results? Must be improved.

I have edited the manuscript to improve the support the results provide for the conclusions.

The abstract is very long and conceptual. Key concepts should be presented, approaches that have been employed, and the expected results to prompt reading continuity. The current formatting makes reading tiresome.

I have revised the abstract by shortening it to present the key concepts, approach and expected results.

In the introduction, the word Overview is unnecessary.

The word “Overview” was removed from the Introduction.

The final chapter in the introduction to the paper organization is missing.

I have reorganize the manuscript by renumbering the Introduction’s sections.

The introduction and its subtopics got very confusing. It is not known where the introduction to the theme is being made and where is the theoretical framework. The reading is confusing.

I have edited the manuscript to separate the introduction to the theme from the theoretical framework to reduce confusion.

Fix typos and formatting on paper.

I have carefully edited the manuscript for typos and formatting.

The symbols of the equations in the paper should be explained. Some of them were not correctly posted.

I have explained all of the symbols in the equations. These comments by the reviewer were particularly helpful.

Honestly, I had a hard time understanding what the main contribution of the paper was. The massive reading of the texts, Lack of schematic images impairs the reader's perception.

I have edited the manuscript to highlight its main contribution of modeling a social harmonic oscillator.  I have added two more images to assist readers.

The article needs to be better explained and highlighted to make it easier for the journal to read interested parties.

I have edited the manuscript to better explain it and by highlighting it to make it easier for interested parties to read.

Reviewer 3 Report

The author presents a conceptual work related to the autonomous human machine teams (HMT). The previous research reported by the author confirms how the maximum interdependence in human teams is associated with the performance of best teams. For example, the research made by the author confirms that the top global oil firms maximize interdependence by minimizing redundant workers. In this paper, the author has generalized this theory by having confirmed that maximum interdependence in teams requires intelligence to overcome obstacles for a team to maximize its entropy production. Thus the author assumes that teams are free to choose their own members, but that freedom, especially for succesful teams, can motivate constructive and destructive interferences (entropy).

The goal of the paper is interesting and intersects the topic of the Journal. I only suggest that the authors should clarify better the main novelty of the work with respect to previous theory. This clarification may help to highlight the impact of the proposed theory and how this theory can be generalized.

Author Response

I have attached my comments in the uploaded pdf. To review, I thanked the reviewer 3 for comments 1-2-3-4-5-6. I also thanked the reviewer 3 for comment 7 and added that I have reworked the manuscript to clarify the main novelty of it with respect to previous theory and to help readers to recognize the theory's impact, its weaknesses, future research and how it might be generalized.   

Round 2

Reviewer 2 Report

Dear author
Thanks for clearing up my doubts.
The article is now more organized and of great quality in its presentation.
Thanks for the improvements.